# Evaluation of a Respiration Rate Sensor for Recording Tidal Volume in Calves under Field Conditions

**DOI:** 10.3390/s23104683

**Published:** 2023-05-12

**Authors:** Lena Dißmann, Petra Reinhold, Hans-Jürgen Smith, Thomas Amon, Alisa Sergeeva, Gundula Hoffmann

**Affiliations:** 1Department Sensors and Modeling, Leibniz Institute for Agricultural Engineering and Bioeconomy (ATB), Max-Eyth-Allee 100, 14469 Potsdam, Germany; 2Institute of Molecular Pathogenesis, “Friedrich-Loeffler-Institut” (Federal Research Institute for Animal Health), Naumburger Str. 96a, 07743 Jena, Germany; 3Research in Respiratory Diagnostics, Bahrendorfer Straße 3A, 12555 Berlin, Germany; 4Department of Veterinary Medicine, Institute of Animal Hygiene and Environmental Health, Freie Universität Berlin, Robert-von-Ostertag-Str. 7-13, 14163 Berlin, Germany; 5System Modeling Group, Institute for Veterinary Epidemiology and Biostatistics, Freie Universität Berlin, Königsweg 67, 10117 Berlin, Germany; sergeea99@zedat.fu-berlin.de

**Keywords:** tidal volume, pressure, respiration rate sensor, spirometry, impulse oscillometry system

## Abstract

In the assessment of pulmonary function in health and disease, both respiration rate (RR) and tidal volume (Vt) are fundamental parameters of spontaneous breathing. The aim of this study was to evaluate whether an RR sensor, which was previously developed for cattle, is suitable for additional measurements of Vt in calves. This new method would offer the opportunity to measure Vt continuously in freely moving animals. To measure Vt noninvasively, the application of a Lilly-type pneumotachograph implanted in the impulse oscillometry system (IOS) was used as the gold standard method. For this purpose, we applied both measuring devices in different orders successively, for 2 days on 10 healthy calves. However, the Vt equivalent (RR sensor) could not be converted into a true volume in mL or L. For a reliable recording of the Vt equivalent, a technical revision of the RR sensor excluding artifacts is required. In conclusion, converting the pressure signal of the RR sensor into a flow equivalent, and subsequently into a volume equivalent, by a comprehensive analysis, provides the basis for further improvement of the measuring system.

## 1. Introduction

Cattle are predisposed to diseases of the respiratory tract due to the morphological and anatomical peculiarities of their lungs [1,2]. This is mainly due to the strong segmentation of the lungs compared to other animal species, the lack of collateral ventilation pathways, and the reduced gas exchange capacity of the lungs, as there are fewer pulmonary capillaries per alveoli unit. In consequence, a larger part of the lungs must be ventilated at rest, which leads to less available ventilatory reserves. Thus, a failure of individual lung areas will result in shortness of breath more rapidly than in other animal species [3]. 

Furthermore, in relation to their body mass, cattle possess a lower total lung volume and a smaller alveolar surface area compared to other animal species. These peculiarities in lung anatomy have significant consequences for pulmonary functions in terms of spontaneous ventilation. For example, a cow of 500 kg has a tidal volume (Vt) of approximately 3800 mL (which corresponds to a Vt of approximately 8 mL/kg body weight), while a horse of a similar weight (550 kg) has a Vt of approximately 6000 mL or 11 mL/kg body weight [4]. Vt is the volume of air inspired and expired during one respiratory cycle, and its measurement allows conclusions about the functionality of the lungs [5]. To compensate for physiologically based lower Vt, cattle must have a higher respiration rate (RR) at rest than other animal species [1,6].

Since bovines do not reach postnatal lung maturity before they have gained a body weight of approximately 300 kg, young animals are particularly susceptible to diseases of the respiratory tract [2]. Therefore, a lot of research has been published on the study of pulmonary diseases in calves and to diagnose them early [7,8,9,10]. Besides the analysis of biomarkers in the blood [11,12], also ultrasonic methods [13,14,15] and different sensors [16,17] have been tested as diagnostic methods. 

To detect pulmonary disorders in calves at an early stage, it is fundamental to monitor the spontaneous breathing pattern. In addition to recording RR (i.e., breaths per minute (bpm)), Vt is a vital parameter in pulmonary function diagnostics. That Vt is a parameter worth measuring in calves has been proven in studies to quantify respiratory symptoms [18,19] or to assess the effects of therapies [20]. It is a well-known phenomenon that calves suffering from pneumonia show significantly lower Vt and higher RR than healthy calves [21,22]. Since Vt is inevitably associated with RR, a higher RR usually compensates for a lower Vt to avoid hypoventilation and to maintain the supply of oxygen to the organism [5].

Several noninvasive methods are available for measuring RR and Vt in animals. RR could simply be counted by observation; however, this requires personnel, time, and thorough quietness in the surrounding to avoid stress for the animals. In 2018, an RR sensor for cattle was developed at the Leibniz Institute for Agricultural Engineering and Bioeconomy that can derive the RR from the pressure difference between inhaled and exhaled breath in one nostril [23]. Most advantageous, this device is suitable for use in freely moving animals under field conditions. Moreover, a complete new non-contact approach to measure RR is infrared thermography using the temperature difference between inhaled and exhaled air to automatically record the RR in calves and adult cattle [24,25]. However, there is not yet a market-ready, universally applicable solution for this method.

Spirometry is another well-known method to assess both RR and Vt. Spirograms can easily be registered in spontaneously breathing animals wearing a tightly fitting facemask (fitting to the size of the animal’s head, thus ensuring low dead space). Since a facemask is mandatory to connect the spirometer in front of the animal’s head, this method cannot be used in freely moving animals. One of the best-validated methods for pulmonary function testing in calves is the impulse oscillometry system (IOS) [26,27]. The system includes spirometric measurements as well. Since the IOS was originally developed for human medicine, it has been applicable to calves with body weights comparable to those of adult humans. Under experimental conditions, IOS has been proven to measure RR and Vt in calves with high accuracy [21,22] (and can therefore be regarded as the gold standard method. The aim of this study was to evaluate whether the previously developed RR sensor could be expanded to measure Vt as well. This new measurement method would offer the possibility to assess the two main variables of the spontaneous breathing pattern continuously in the field without the necessity of connecting a facemask or fixation of the animals to be monitored. According to our conceptual goal, the high frequency pressure values (50 Hz) measured by the RR sensor should be converted into airflow and volume equivalents to ultimately compare them to the real flow and volume measured by the IOS. The physiological Vt for calves is approximately 8–10 mL/kg [6,11]. This proof-of-principle study focused on calves since IOS measurements are limited to bovines with body weights below 120 kg. 

The paper is structured as follows: Section 2 and Section 3 describes the methodology as well as the statistical analysis of the sensor data. Section 4 presents the results. Afterward, the results are discussed in Section 5. Finally, the conclusion is reported in Section 6. 

## 2. Animals, Materials and Methods

### 2.1. Study Design

The animal testing took place at a research farm (Groß Kreutz, Germany) on two consecutive days in April 2022. The testing was approved by the State Office for Occupational Safety, Consumer Protection and Health (LAVG Brandenburg, Potsdam, Germany) under the study number: V6-2340-12-2022 (day of permission: 25 February 2022).

Selection criteria for the calf’s inclusion in the testing were clinical health and a maximum weight of 80 kg, since the mask does not fit larger animals. Clinical health was ensured by a veterinarian before the habituation process and repeated before animal testing. The clinical examination included the measurement of rectal temperature, examination of the lymph node and mucosal status, capillary refill time, lung and heart auscultation (including determination of RR and heart rate), and assessment of general condition. Ten clinically healthy calves (aged between 4 and 65 days on the first day of the experiment) were included. Using a sample size estimation tool [28], a case number of 10 calves was necessary to prove a correlation of at least 0.8 [29] of the respiratory parameters between the sensors with an α level of 0.05 and a power of 80%. These α and β levels are commonly used in other clinical studies as well [30]. Body weight ranged from 52 to 79 kg. Three weeks before the testing, the habituation process began (with the exception of the 2 calves that were 4 days old at the start of the testing) while each calf was equipped with the facemask once per week for 5 min. Measurements of spontaneous breathing were performed using two devices on each calf in a randomized order. On Day I, measurements with the RR sensor preceded IOS-measurements and vice versa on Day II. All measurements were performed in standing animals with a standardized, slightly stretched head position. Ambient conditions on the two consecutive days were comparable (Day I: ambient temperature 10 °C, relative humidity 51%; Day II: ambient temperature 11 °C, relative humidity 55%). 

### 2.2. Measurements with the RR Sensor

First, the calf to be measured was equipped with a foal halter to which the power bank and the RR sensor were attached and connected over a USB cable (Figure 1). The dimensions of the RR sensor with the microcontroller (Gouna, Brandenburg, Germany) were 46 mm × 15 mm × 25 mm (length × width × height). The RR monitor included a sensor, microcontroller, and silicon tube, with a total weight of 45 g, plus a power bank (capacity 2600 mAh) weighing 60 g (Varter Consumer Batteries GmbH, Ellwangen, Germany). The power bank allows continuous recording of the pressure for about 6 h. However, the RR sensor is not waterproofed, so it must be removed after a short measurement period [23]. A flexible silicone tube with an inner diameter of 2 mm lead from the sensor to the nostrils where it was fixed with a nose ring and intruded 1 cm into the nasal cavity of the right nostril to transmit the recorded pressure to the RR sensor (Figure 1). The pressure was recorded continuously during inspirations and expirations over a period of 5 min per calf and the data were stored on a secure digital memory card. Since it was the first measurement with the RR sensor in calves, there were no prior comparable studies, which could have been used as a guide regarding the length of the measurement. The aim was to measure as close in time as possible with both measuring devices, but still collect valid measurement data. Experiences from individual test measurements before the actual testing have shown that 5 min is sufficient to collect valid data from the calves. Afterward, the device was cleaned with particular attention to ensure that the tube of the RR sensor was free of contamination from the inside before each new test to avoid possible measurement distortions due to liquids in the tube (e.g., from sneezing).

### 2.3. Measurements with the IOS

The application of the IOS to the head of a calf is shown in Figure 2. The technical design of this device and the careful evaluation of pulmonary function testing in this animal species have been described elsewhere [26,27]. For IOS data, a heated pneumotachograph (Lilly-type, mesh resistance: 36 Pa L^−1^ s) connected to a differential pressure transducer (SensorTechnics SLP 004D, Puchheim, Germany) ensured continuous measurement of airflow. At flow rates < 15 L s^−1^, the pneumotachograph is linear within 2%. Volume was gained from the flow signal by calculating an integral from airflow during each inspiration or expiration. The results are given as volume curves over time (Figure 3). Three consecutive measurements were registered per calf per day (each included at least 10 regular breathing cycles free of artifacts) lasting on average 24 s per measurement. The number of measurements was based on previous performed studies with the IOS in calves [21]. Read-out variables were: RR (IOS), Vt (IOS), and the minute volume of respiration (Vmin (IOS)) calculated with the formula:Vmin(IOS)=RR(IOS)×Vt(IOS)

### 2.4. Data Evaluation (RR Sensor) 

First, we selected sequences of the recorded 5 min periods where the calves had breathed regularly and evenly for at least 5 breaths in succession (constant frequency and constant amplitude). Since one dataset of the RR sensor from one calf was not readable, 19 datasets from both devices were included. At least 2 to a maximum of 5 sequences of the RR sensor per calf and day were selected for the evaluation, resulting in a total of 59 selected periods included in the evaluation.

Subsequently the pressure data were adjusted according to an exponential smoothing formula [31]. Afterward, the inhalation cycle was adjusted to body temperature and pressure saturated (BTPS) conditions converting inhaled pressure samples measured under ambient conditions to the condition in the lung. This kind of conversion is required to exclude measurement distortions due to ambient temperature and humidity [32]. Consequently, the obtained pressure represented the flow equivalent, with exhalation generating positive pressure and inhalation generating negative pressure (Figure 4).

Second, we integrated the pressure values to obtain the volume equivalent [33] (Figure 5). In addition, a baseline correction was performed to compensate for the shift of the zero point (Smith, personal communication) which is caused by an insufficient BTPS-approximation. 

Afterward, the difference between the maximum and minimum pressure of each in- and exhalation was calculated representing the Vt equivalent (Figure 6). Subsequently, since the data were not normally distributed, the median of all Vt equivalents for each of the 59 selected periods was calculated for the RR sensor. Moreover, the RR for each data period as well as the respiratory minute volume (Vmin) equivalent was calculated by multiplying the Vt equivalent by the RR.

## 3. Statistical Analysis 

The dataset consisted of two different sensor systems, on the one hand the IOS as the gold standard and on the other hand the RR sensor as the new application method. For the statistical evaluation, we analyzed three parameters of each sensor system: RR, Vt, and Vmin. In total, we had 59 selected sequences for the RR sensor (2–5 measurements per calf and day) and 54 observations for the IOS (2–3 measurements per calf and day). We tested the raw variables for normality using the Shapiro Wilk test from “scipy” python3 package and the alpha level was selected as 0.05 [34]. Since most parameters (except Vt equivalent (RR sensor)) were not normally distributed, the median, minimum, as well as maximum and quartiles were calculated for each parameter. 

### 3.1. Sensor Calibration

Since the RR sensor uses a different algorithm of measurement, which likely causes a systematic mistake in the results, the systematic mistake was removed by performing a sensor calibration. This mistake can be easily found in the plots with raw results (Figure 7, Figure 8, Figure 9, Figure 10, Figure 11 and Figure 12). The observations, taken from the same animal, however, were slightly moved along one of the axes (along RR or IOS sensor), which gave an idea of a systematic mistake in the device. Therefore, an individual correction factor based on the mean values across the two days of observation was calculated to remove the systematic mistake [35]. For example, if the mean values across one calf were equal to 25 bpm in the RR sensor and 27 bpm in the IOS sensor, for this calf, the correction factor was 25/27, which is approximately 0.93. Then, in all the RR observations from the RR sensor, we would multiply the raw RR values by 0.93, and this new value is calibrated. We applied the same procedure to each measured parameter. 

The formula below describes the calibration process, where x is the RR sensor observation, μRR is the mean value of the RR sensor parameter, and μIOS is the mean value of the IOS parameter:$$correction\;factor=\frac{x\cdot \mu \mathrm{IOS}}{\mu \mathrm{RR}}$$

After the calibration was done, the plots started to show some correlation pattern, which were further analyzed.

### 3.2. Z-Score Transformation

Due to a different unit system in the sensors, we faced the problem of measurements incompatibility and applied Z-score transformation (standardization) to all measurements. We calculated the mean value and standard deviation for every measured parameter separately (RR, Vt, and Vmin) and transformed the raw values to the Z-scores, so that 95% of observations were placed between −2 and +2. That helped comparing the values in the Z-Score unit system [36].

The Formula below describes the Z-score standardization (x_z_), where x is the observation value, μ is the mean value of the parameter, and σ is the standard deviation of the parameter:$$x_{z}=\frac{x-\mu }{\sigma }\;$$

### 3.3. Normality Testing

To decide which statistical approach (parametric or non-parametric tests) to use, we also tested the calibrated variables for the normality using the Shapiro Wilk test from “scipy” python3 package [34] and selected α level as 0.05. Since only the calibrated Vt (RR sensor) was normally distributed, we chose the non-parametric approach further.

### 3.4. Non-Parametric Test

Due to lack of normally distributed parameters, we used two-tailed Wilcoxon and two-sample Kolmogorov–Smirnov tests to investigate whether the observations from two sensors (RR sensor and IOS) were taken from the same distribution or not [37,38]. Therefore, the null hypothesis was that the observations had the same source. The alternative one was that they had different origins. 

We analyzed all values we had (standardized by z-score transformation 54 observations for the IOS and 59 for the RR sensor) from all calves and both days. We compared the parameters as pairs (RR, Vt, and Vmin from both sensors), resulting in 46 observations for every parameter; after removing not completed observations, we also compared both raw and calibrated values. The two-tailed Wilcoxon and two-sample Kolmogorov–Smirnov tests were performed using “scipy” pyhton3 package with basic parameters. 

The α-level was selected using Bonferroni correction from “statmodels” python3 package with “not-sorted” setting, and it was 0.0167 [39]. 

### 3.5. Correlations

Since the data were not normally distributed (see Section 3.3 Normality Testing), we performed the Spearman correlation test using “scipy” python3 package for both raw and calibrated data. In addition, we performed the Spearman test for medians per animal and medians per animal per day [38].

Moreover, we also analyzed the correlation between Vt and RR for each sensor separately to analyze whether the sensor systems showed the same characteristics with changing RR.

## 4. Results

The calf’s recorded RR was similar for both devices ranging between 19 and 82 bpm, while the Vt of the IOS varied from 6.64 mL/kg to 14.29 mL/kg body mass and the Vt equivalent of the RR sensor ranged from 6 to 60 (an equivalent is unit-free) (Table 1). After the calibration process, the range of the RR sensor values was closer to the values of the IOS and therefore, better comparable to each other.

Since the calibration coefficients, calculated from the median per calf of the RR sensor and IOS, were inhomogeneous between the single calves (RR: 0.66–1.27, Vt: 19–71 and Vmin: 0.02–0.06), we used the median in the following statistical evaluation.

The two-tailed Wilcoxon test did not show a significant difference neither between raw values nor between calibrated values (Table 2). Therefore, we cannot reject the null hypothesis, that both sensors took their measurements from the same distribution. The Kolmogorov–Smirnov test was used as a second method, which also did not show any significant difference between RR and IOS sensors (*p*-value for all parameters > 0.8). The Wilcoxon test showed that the differences inside the groups (RR sensor and IOS) did not significantly differ from the differences between groups, which can partly show that the sensor observations were taken from the same distribution.

After applying the Bonferroni correction, our conclusion did not change because our *p* values were much higher than α level. 

A correlation between the raw data of both measurement systems was not visible (Spearman correlation coefficient (CC): 0.1~0.37). However, once we added the information about systematic mistakes in the RR sensor (calibration), the correlation in RR and Vt became clearer across the whole dataset without extracting medians from them (Table 3, all observations, Spearman CC: 0.2~0.48). The most obvious correlations were found in medians per animal (Spearman CC > 0.9, *p*-value < 0.05 in RR and Vt). However, in Vmin, the *p*-value did not provide statistical significance (*p*-value > 0.05) [38]. 

In summary, a strong correlation (Spearman CC > 0.9) between the sensors was only achieved for the median per calf of RR and Vt, but not when comparing all raw observations [29].

In addition, we investigated the Spearman CC between Vt and RR for both sensor systems separately, to analyze if they show the same characteristics. It is already known from the literature that a high RR (shallow breathing) is associated with a lower Vt and vice versa [5]. As demonstrated in Table 4, there was a significant negative correlation for the raw and for the calibrated RR sensor data as well as for the IOS data. Thus, the Vt equivalent showed the same characteristics with changing RR like the Vt (IOS) since the Vt equivalent increased with decreasing RR. 

## 5. Discussion

The primary objective of this study was to investigate whether a valid Vt parameter could be extracted from the pressure signal measured by the RR sensor. Our results indicate that a flow equivalent as well as a Vt equivalent could be derived from the pressure (RR sensor). In addition, a high correlation between the calibrated RR (RR sensor) and the RR (IOS) as well as between the calibrated Vt equivalent (RR sensor) and the Vt (IOS) for the median per calf was shown. However, the Vt equivalent could not be converted into a true volume given in mL or L because the correlation between the raw data when facing all observations of both measurement systems was too low [29].

In summary, this study showed in detail how the RR sensor can be used in calves and how the pressure can be converted into a Vt equivalent. The Vt equivalent performed simultaneously with the Vt (IOS) since both are negatively correlated with the RR. The study can serve as a basis for future research in the field of lung function diagnostics in calves. Furthermore, it also points out the necessary technical improvements that have to be performed on the RR sensor to reliably record the Vt. If these technical adaptions were implemented, it would also be possible to develop such a diagnostic tool for adult cattle as well. At the current state of the art, the Vt of adult cattle can only be determined with a facemask [40] or estimated by equations from respiratory data [41] in order to detect heat stress at an early stage. Measuring the real Vt in adult cattle without a facemask would be an improvement in the lung function diagnostic. Since the study took place under real field conditions and was the first attempt to determine the Vt using an RR sensor, it would be the first RR sensor capable of measuring the two main respiratory parameters in freely moving animals under field conditions without restrictions on normal behavior. 

Compared to assessing pulmonary functions by using the IOS or other spirometers in individual animals, the RR sensor can be worn for a longer period without the need for a prior habituation process. Furthermore, simultaneous recording of RR and Vt would be possible in a larger number of animals due to less personnel and resource requirements. This non-invasive approach of data recording is also an advantage over other methods of lung examination, such as ultrasound [13,14,15] and blood sampling [11,12], where a fixation of the calf is mandatory. The IOS was used as the gold standard because (i) it had previously been thoroughly evaluated for pulmonary function testing in calves, and (ii) it is suitable for the assessment of spirometric parameters of spontaneous breathing. Respiratory impedance as the main measure provided by IOS was not taken into account in this study, which focused on the measurement of RR and Vt. In the present study, the measured Vt values of the calves were within the physiological range since the reference values for calves are approximately 8–10 mL/kg [6,21].

### Limitations and Future Research 

Despite the habituation process, the calves were partially nervous during the experiment, as evidenced by hyperventilation (RRs up to 82 bpm), taking into account that the physiological RR for calves ranges from 20 to 30 bpm [5,18]. Therefore, a simultaneous measurement of the two devices would be useful to allow a direct comparison between the continuous readings as well as a longer habituation period. Unfortunately, simultaneous measurement with both devices was not possible in our animal testing due to the lack of space under the facemask for the RR sensor. 

Furthermore, unsteady movements of the calves as well as moisture in the hose of the RR sensor (caused by sneezing) led to an influence of the pressure because the membrane of the RR sensor reacts very sensitively to changes in the environmental conditions. As a result, it became partly difficult to distinguish between physiological recorded pressures and artifacts. 

In addition, we determined that reliable measurements with the RR sensor are only technically possible with calves from approximately 1 week of life onwards. The fixation of the nose ring in very young calves (up to approximately 4 days old) was not optimal, since it slips easily which also influences the pressure. Furthermore, the membrane of the RR sensor is subject to pressure fluctuations of 0.06%, which corresponds to approximately 0.03 mbar, thus the pressure can never be recorded with 100% accuracy [42].

The statistical results indicate that the RR sensor can replace the IOS in the future. Since the data showed a high correlation between the RR sensor and IOS for the calibrated RR and VT for the median per calf, the RR sensor seems to be a promising replacement. Regardless, the two-tailed Wilcoxon and Kolmogorov–Smirnov tests showed no significant difference between the two sensors; we did not have enough evidence to talk about sensors replacement validity yet because only the median per calf showed a high correlation between the two sensors. Therefore, at least 2–5 observations per calf of the RR sensor as well from the IOS were necessary to find the calibration coefficient, which provided a high Spearman CC and good significance. 

Moreover, to replace the gold standard method, it is necessary to find the accurate algorithm, which would recalculate the results of the RR sensor in the unit-free system into mL. Therefore, it must be taken into account that the data in our study were very specific for this age and breed, as it is influenced, among other factors, by the diameter of the nostrils. For an accurate and valid approximation of the calves’ Vt across all breeds and ages, it is necessary to expand the number of cases with calves differing in age, breed, weight, and environmental conditions.

However, a prior revision of the RR sensor would be required to reliably record the pressure without artifacts and to finally achieve a possible higher correlation in the raw observations. Therefore, the following technical improvements to the RR sensor would be necessary: an installation of a position sensor to ensure that only measurements are recorded when the calf is at rest and does not make restless head movements. This technical adaption would have the advantage that artifacts would automatically be absent. In addition, it would be useful to be able to record the pressure in both nostrils to exclude pressure changes due to calves’ movements and pathological unilateral nasal constriction due to fluid accumulation. Moreover, it would also contribute to the precision of the data (as well be a huge reduction in workload) if the evaluation steps that we have manually performed to convert the pressure into a Vt equivalent were automatically integrated by the RR sensor. This would practically mean to automatically take into calculation the BTPS correction based on temperature and humidity measurements, smoothing of the data, as well as automatic integration of the pressure values and finally calculating the difference between maximum and minimum pressure in each breath.

Furthermore, a waterproof cover with a longer battery life (currently approximately 6 h) would be useful, enabling the RR sensor to be worn during drinking and thus over an even longer period of time. There is already a market-ready RR sensor from Gouna [43] available, which is water-resistant and allows RR measurement in cows over approximately 6 months. However, this is only adapted for cows and would need to be made smaller and lighter for calves. In addition, the market ready RR sensor only records the RR and no longer outputs the individual pressure differences, which is why it is not suitable for recording the Vt equivalent without complete revision. 

The outlined technical adaptions of the RR sensor would essentially simplify the measurement method of Vt for the future because the RR sensor does not impair the normal behavior of the animals (no mask is required) and can be used at all ages from young calves to adult cattle.

In summary, it can be stated that before further animal testings are performed, a technical revision of the RR sensor is necessary (in particular: an installation of a position sensor, a waterproof cover, measurements in both nostrils, and a longer battery life). Moreover, the development of a larger facemask would be required to install the RR sensor under the facemask and therefore allow a simultaneous measurement with both devices. If these technical improvements were implemented, a testing on calves differing in age, breed, weight, and environmental conditions would be useful to develop a reliable algorithm converting the Vt equivalent into volume in mL.

## 6. Conclusions

This study demonstrated how to convert the pressure of the RR sensor into a flow equivalent and subsequent volume equivalent by a comprehensive analysis. It could be shown that the pressure (RR sensor) behaved synonymously with the flow (IOS) and that the median of the calibrated Vt equivalent of the RR sensor per calf showed a high correlation with the Vt of the IOS. Thus, the RR sensor appears to be fundamentally suitable as a measurement parameter of the Vt. However, the Vt equivalent (RR sensor) could not be converted into a true volume in mL or L because the correlation between the raw data was too low. This proof-of-principle study provided the basis for further research focusing on technical adaptations of the RR sensor to determine reliable Vt data in parallel to RR and to develop a market-ready device. 

## Figures and Tables

**Figure 1 sensors-23-04683-f001:**
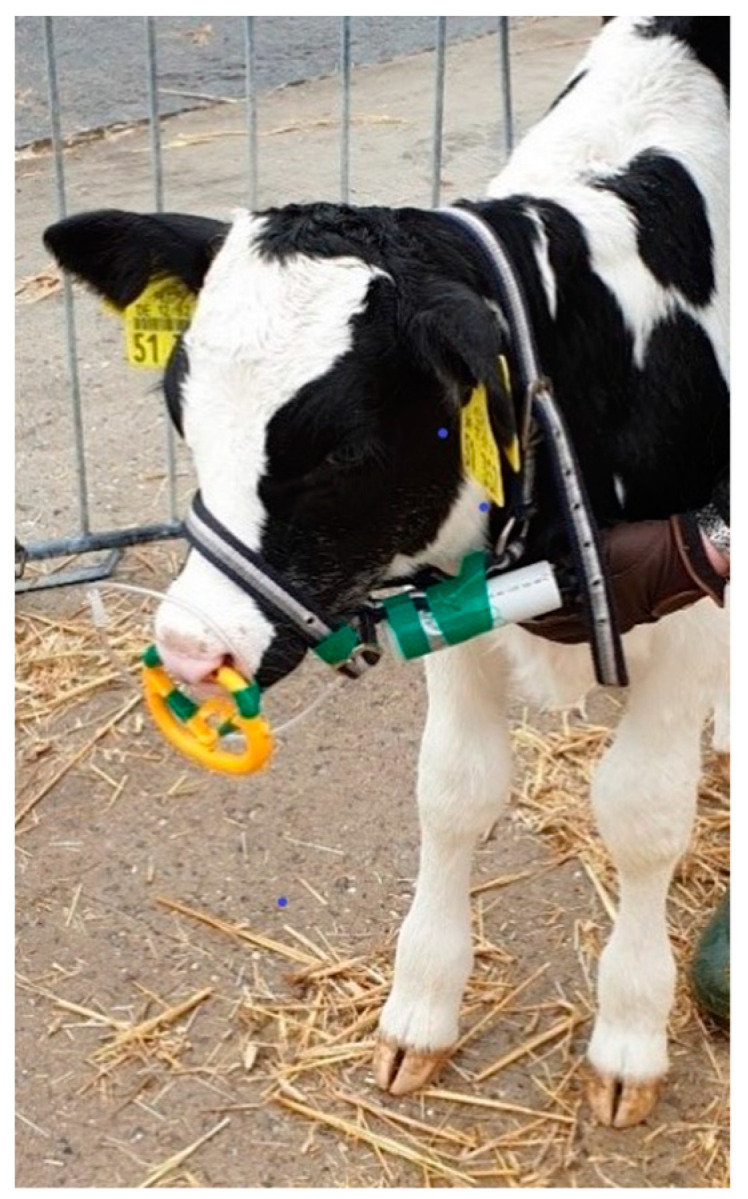
Respiration rate sensor fixed on the calf’s nostrils with a nose ring.

**Figure 2 sensors-23-04683-f002:**
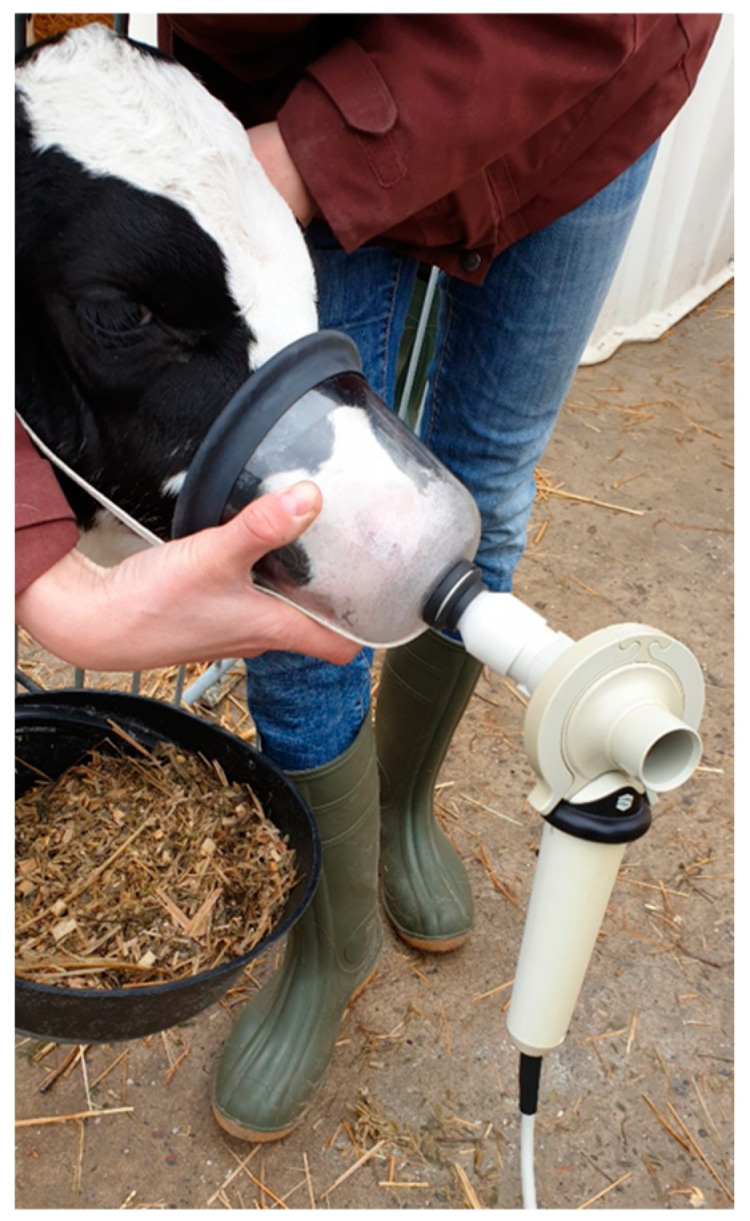
Impulse oscillometry system adapted to a tightly fitting rigid facemask.

**Figure 3 sensors-23-04683-f003:**
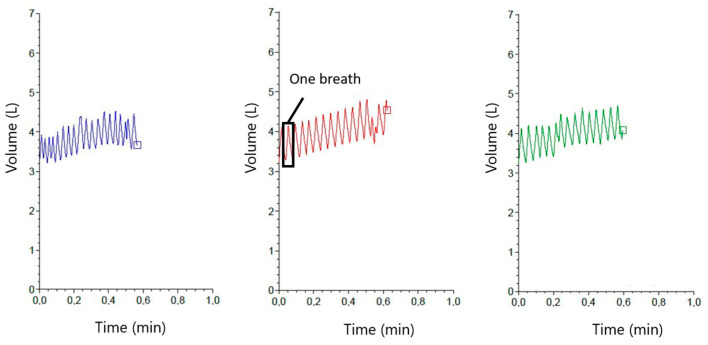
Impulse oscillometry system (IOS): Three consecutive measurements (each measurement is represented by a different color) of the tidal volume (difference between maximum and minimum of the Volume in L) and the respiration rate (in breaths per minute) of one calf registered by the IOS.

**Figure 4 sensors-23-04683-f004:**
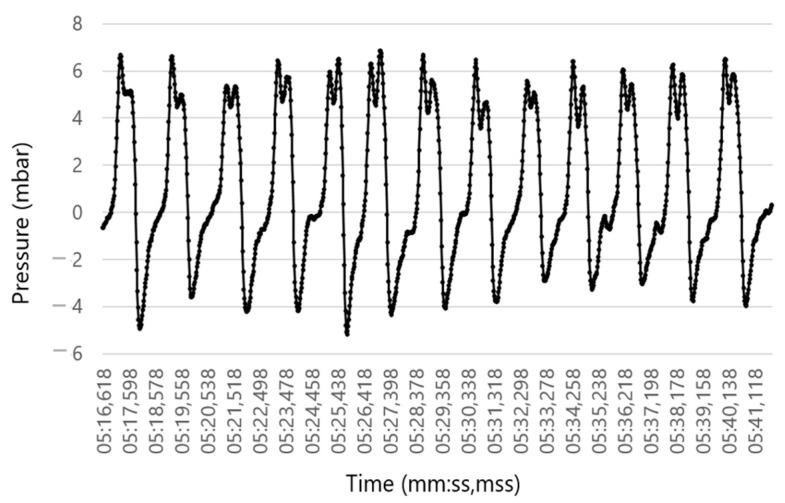
Respiration rate (RR) sensor: Pressure registration of the RR sensor corresponding to the flow equivalent after smoothing and adjusting the inhalation cycle to body temperature and pressure saturated (BTPS) conditions. Exhalation is associated with a positive pressure and inhalation with a negative pressure.

**Figure 5 sensors-23-04683-f005:**
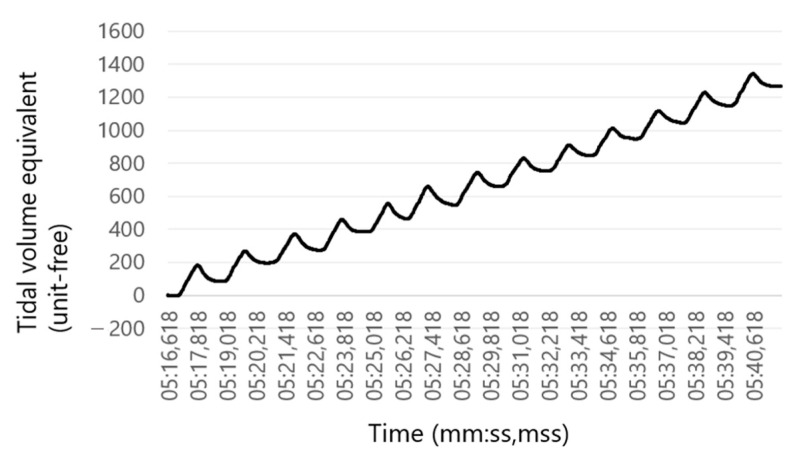
Respiration rate sensor: Integrated pressure curve corresponding to the tidal volume equivalent.

**Figure 6 sensors-23-04683-f006:**
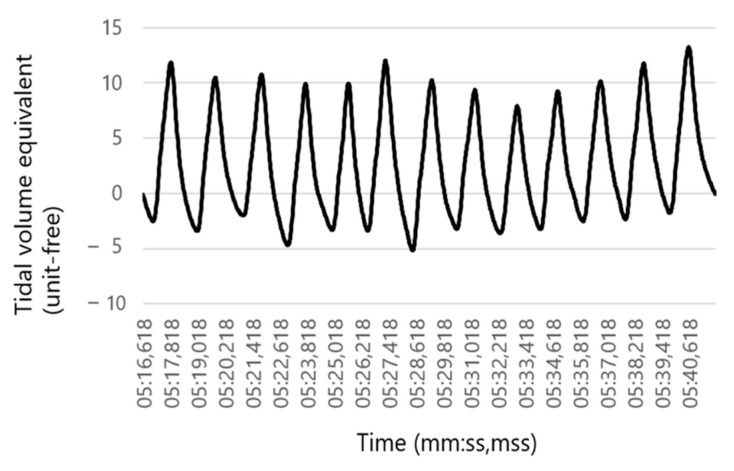
Respiration rate (RR) sensor: Tidal volume equivalent of the RR sensor after baseline correction.

**Figure 7 sensors-23-04683-f007:**
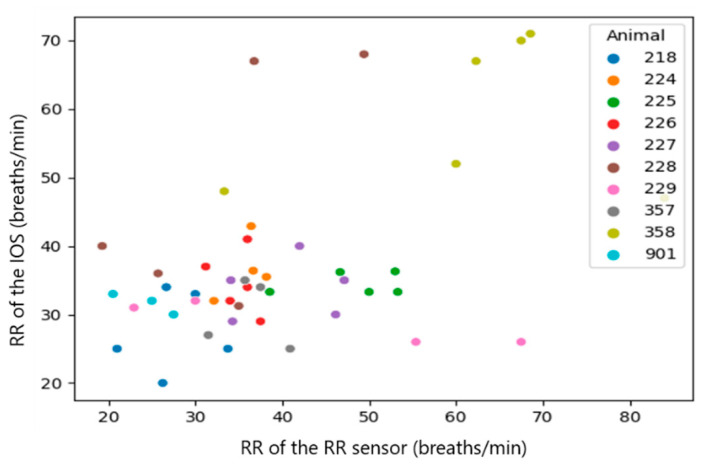
Scatter diagram showing the raw respiration rate (RR) measured by the impulse oscillometry system (IOS) and the RR sensor (*n* = 46). The data of the individual calves are marked with different colors.

**Figure 8 sensors-23-04683-f008:**
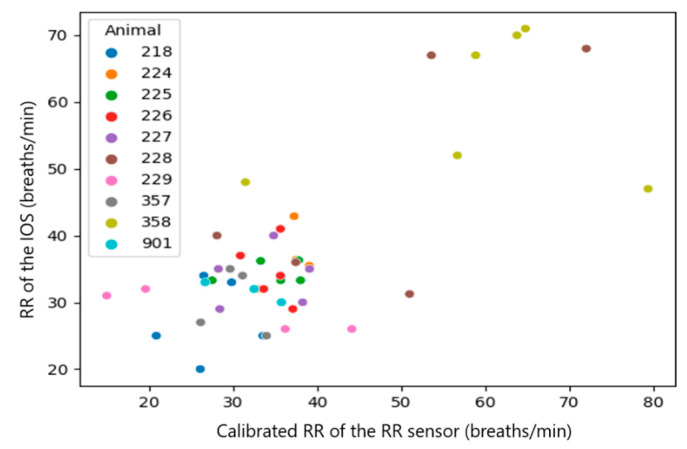
Scatter diagram showing the calibrated RR measured by the IOS and the RR sensor (*n* = 46). The data of the individual calves are marked with different colors.

**Figure 9 sensors-23-04683-f009:**
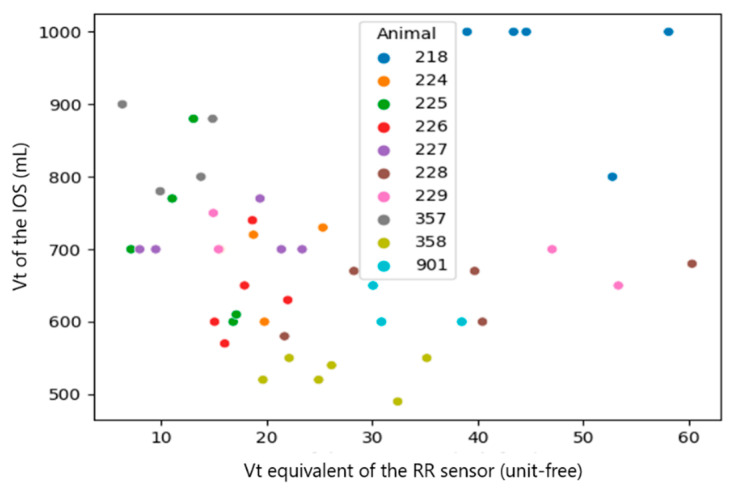
Scatter diagram showing the tidal volume (Vt) measured by the IOS and the raw Vt equivalent of the RR sensor (*n* = 46). The data of the individual calves are marked with different colors.

**Figure 10 sensors-23-04683-f010:**
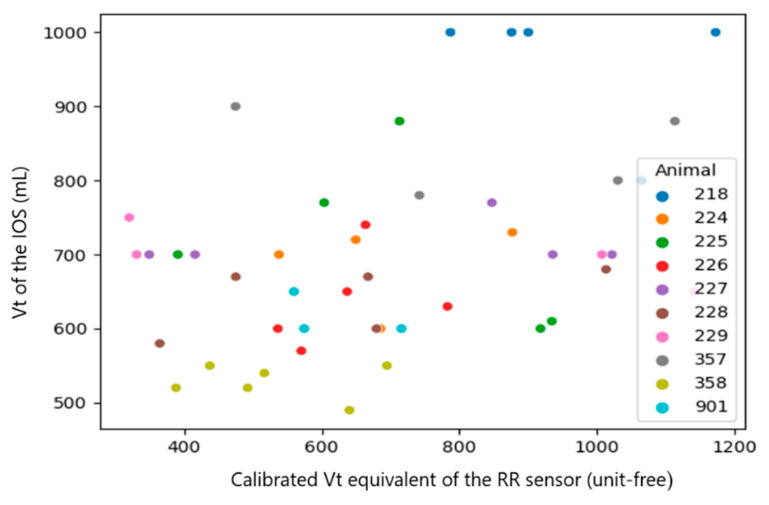
Scatter diagram showing the Vt measured by the IOS and the calibrated Vt equivalent of the RR sensor (*n* = 46). The data of the individual calves are marked with different colors.

**Figure 11 sensors-23-04683-f011:**
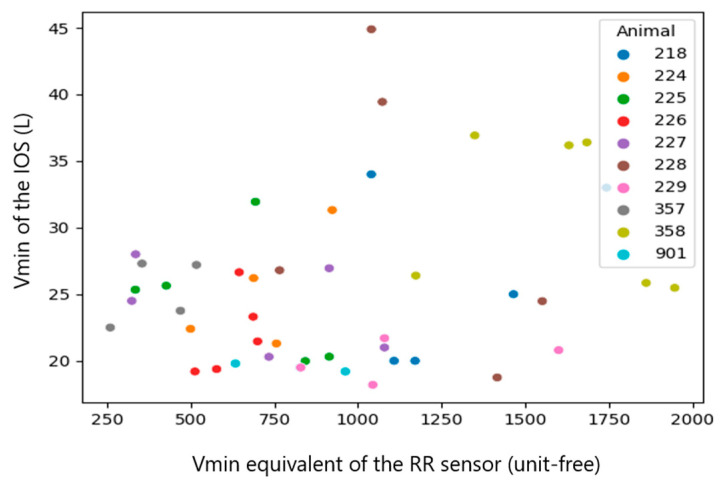
Scatter diagram showing the minute volume (Vmin) measured by the IOS and the raw Vmin of the RR sensor (*n* = 46). The data of the individual calves are marked with different colors.

**Figure 12 sensors-23-04683-f012:**
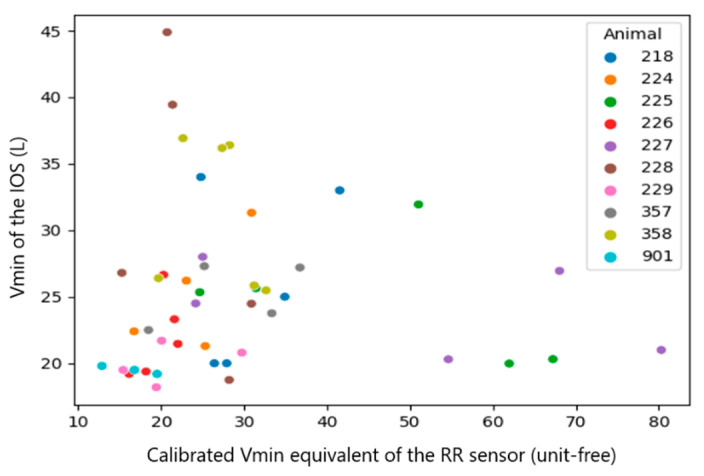
Scatter diagram showing the Vmin measured by the IOS and the calibrated Vmin of the RR sensor (*n* = 46). The data of the individual calves are marked with different colors.

**Table 1 sensors-23-04683-t001:** Raw parameters of respiration rate (RR), tidal volume (Vt), and minute volume (Vmin) of RR sensor (*n* = 59) and impulse oscillometry system (IOS) (*n* = 54).

Parameter	Unit	Min	0.25	Median	0.75	Max
RR (RR sensor)	breaths per min (bpm)	19	30	36	47	82
Vt equivalent (RR sensor)	unit-free	6	15	19	30	60
Vmin equivalent (RR sensor)	unit-free	260	529	762	1079	1946
RR (IOS)	bpm	20	31	34	40	71
Vt (IOS)	mL	425	600	700	773	1000
Vmin (IOS)	L	18	21	24	27	45
Vt/kg (IOS)	mL/kg	6.64	7	11.11	12.38	14.29

**Table 2 sensors-23-04683-t002:** *p*-values of the Wilcoxon test for raw and calibrated parameters between respiration rate (RR) sensor and impulse oscillometry system (IOS) of all observations (*n* = 46).

Parameter	*p*-Value	Difference
Raw RR (RR sensor) and RR (IOS)	0.7291	No
Raw Vt (RR sensor) and Vt (IOS)	0.9914	No
Raw Vmin (RR sensor) and Vmin (IOS)	0.9655	No
Calibrated RR_(RR sensor) and RR (IOS)	0.8596	No
Calibrated Vt and Vt (IOS)	0.8288	No
Calibrated Vmin and Vmin (IOS)	0.768	No

Vt = tidal volume; Vmin = minute volume.

**Table 3 sensors-23-04683-t003:** Spearman correlations between respiration rate (RR) sensor and impulse oscillometry system (IOS) for all observations (*n* = 46) and median per calf (*n* = 10) after calibration process.

Parameter	Number of Cases	Spearman Correlation	*p* Value
RR_calibrated (RR sensor) and RR (IOS)	all observations (*n* = 46)	0.4813	0.0007
	Median per calf (*n* = 10)	0.9515	0.000022799
Vt_calibrated (RR sensor) and Vt (IOS)	all observations (*n* = 46)	0.3709	0.0112
	Median per calf (*n* = 10)	0.9142	0.0002
Vmin_calibrated (RR sensor) and Vmin (IOS)	all observations (*n* = 46)	0.2246	0.1334
	Median per calf (*n* = 10)	0.5636	0.0897

RR = respiration rate; Vt = tidal volume; Vmin = minute volume.

**Table 4 sensors-23-04683-t004:** Spearman correlation between respiration rate (RR) and tidal volume (Vt) of the impulse oscillometry system (IOS) (*n* = 54) as well as between RR and Vt equivalent of the RR sensor (*n* = 59).

Parameter	Spearman Correlation	*p*-Value
RR and Vt equivalent (RR sensor)	−0.513	0.0003
Calibrated RR and Vt equivalent (RR sensor)	−0.4621	0.0012
RR and Vt (IOS)	−0.5438	0.000094007

## Data Availability

The data that support the findings of this study are not openly available due to reasons of sensitivity but are available from the corresponding author upon reasonable request.

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
