# Peer review of "Evaluation of a Respiration Rate Sensor for Recording Tidal Volume in Calves under Field Conditions"

_sensors, 2023, doi:10.3390/s23104683_

Round 1
Reviewer 1 Report
In this paper, the Authors, in their own words, aim to 'evaluate whether a RR sensor which was previously developed for cattle is suitable for additional measurements of Vt in calves'.
The document is well written, without relevant English language issues. It is organized as follows: Abstract, a first Section containing an Introduction, a second Section on Animals, Materials and Methods, a third Section on the Results, a fourth Section with the Discussion, a fifth Section containing the Conclusions of the work, and finally the References used.
After a detailed review, I must say about the document that I think it might in principle result of some interest, but its claimed contributions might not be supported by the shown results. Therefore, I think that some clarifications are needed in order to properly understand the proposal and appreciate its contributions. Please, see my comments below.
1. The proposed title seems not to clearly convey the nature and goals of the work: 'evaluate whether a RR sensor which was previously developed for cattle is suitable for additional measurements of Vt in calves', which is way more specific than the current title: 'Evaluation of a sensor based method for recording tidal volume in conscious calves under agricultural conditions'.
2. Taking into account its key role in the document, the RR sensor used should be described in more detail, with special attention to the aspects that are more relevant for the work, including information about its appearance, placement on the animal, characteristics, technologies, data collection specifics, how it is powered, weight, size, etc.
3. At the end of the Introduction section, a short description about the structure of the document is recommended to inform the reader about how it is organized and what to expect in the next sections.
4. In Section 2.1, some explanation about the criteria for selecting the number and age of the animals, the length of the study and other study data is expected. Same can be said for the experiement settings and measurements described in Sections 2.2 and 2.3.
5. Mathematical expressions in Lines 149-150 could be written as equations, explaining the variables and their units in a way that is easier to read and to understand.
6. In Tables 2 and 3, the second and third rows are exactly equal, is this correct?
7. Please explain better the meaning of 'decreases hyperbolic with increasing factor' in Line 239. In the same line, elaborate some more the explanation in the description of Figure 9.
8. The sentence in Lines 242-243 seriously questions the contributions of this work. I would suggest to address this issue in depth in the Discussion Section.
9. I consider necessary to extend the Discussion section, which must present the relevance and the impact of this proposal in the field of study. Why is this better than any other already existent technologies? What of it is better/worse? Also, it must support the claims made in the Conclusions section.
10. How do the Authors support the claim made in Lines 288-290? What is the logic that leads you to that conclusion? Please, explain.
11. The caveats shown in lines 291-295 severely affect the value of the contributions of this work, even more as some of the factors mentioned were not addresed at all in the previous sections.
12. The Conclusions are too short and, in my opinion, not supported by the contents of the Results and Discussion sections.
13. From a practical point of view, what could be the cost and work needed for implementing a system like this in a real farm? How would be the data recovery from each animal and the device battery charge made? What would be the investment needed?
14. I also have some additional questions about the practical application of this proposal, and its possible use in the real agricultural environments, regarding device robustness against animal movement and behavior, resistance to environmental agents, bio-contamination, long-term negative effects on animals' bodies, etc.
15. The presence in the References of 11 works by one of the Authors (P.R.), out of a total of 21, is very worrying as a potential sign of unnecesary self-citation. Please, justify this need for those references, as well as the lack of need of independent sources about the topics addressed in the manuscript.
16. As a consequence of the above considerations, I would think this document needs additional experiments and studies in order to make a relevant contribution to this field of study, and I encourage the Authors to proceed along that way, elaborating and submitting a new version of the manuscript addressing said considerations.
17. Other comments
• Line 44: There seems to be an extra space before 'To compensate'.
• Tables 1 and 2: Please use commas in '2,269'.
• Lines 217-218. Check the syntax of 'in per calf and day'.
• Table 2: Please correct 'volume/Kg' to 'volume/kg'.
• Line 270: There seems to be an extra space before 'Consequently'.
• Line 287: Please correct 'First sensor 2022' to 'First sensor, 2022'.
• Lines 330-339 should be removed, perhaps.
Reviewer 2 Report
The article is interesting and very well written. Importantly, it raises the current and important topic of preventing respiratory diseases in calves. My main reservation concerns the methodology - the authors compare the measurement results using the reference method (IOS) and their own method (Respiration rate sensor, RR sensor). Although they measure on the same animal, they compare the results obtained in a time interval (first IOS then RR sensor, or first RR sensor then IOS). Hence my question: Since the RRS is similar to a nose ring and connects with a flexible tube to the receiver connected to the halter, is it possible to measure RR and Vt simultaneously with IOS and RRS? Such results would be much more reliable. If not, please discuss this issue in the discussion section as a limitation of this research.
Unfortunately, the Statistical analysis section is described incorrectly and needs to be corrected. It would be advisable to compare the values of the three variables (RR, Vt, and Vmin) using an appropriate statistical test and demonstrate no differences or differences between data series. Then it would be advisable to calculate the correct correlation coefficient (see detailed comments) for the three variables in question. There is no room for guesswork in the Statistical analysis section. If we are talking about a normal distribution or fluctuation of the data, they have to be measured/demonstrated. I'm sure you didn't count one single factor/coefficient to convert the VT equivalent. Why is the one shown in Figure 9 the best? It would be advisable to support the choice of factor/coefficient with curve-fitting indicators.
Detailed comments regarding the following lines of the article are provided below.
Title: Is the fact that the calf is conscious crucial for the measurement of measurement innovation? if not consider removing that word from the title. Consider also changing the title and throughout the whole manuscript body from "agricultural conditions" to "field condition", as it was stated in L 66.
Abstract: The abstract should be rephrased - the results should be included. Concerning the limitation of the abstract section lengths, one may noted that sentence in L 16-17 can be removed and the next one can be shortened.
Manuscript body:
L 44 remove the additional space mark.
L 44 use the abbreviation (Vt) consequently throughout the whole manuscript body.
L 55 Considering changing "have" to "manifest" or "show"
L 57 Could you provide the research on the relation (calculated using correlation coefficient or regression model) between RR and Vt in cattle (including calves) or in any other species?
L 72 However, spirometry can be used in conscious animals. Thus again, consider the importance of the fact that the animal is conscious.
L 75 (Reinhold, in press) I am not sure if such kind of reference is acceptable. Please add a repository in references to this item (biorxive, arxive, or any other) where the reader can consult the preprint if you want to cite unpublished work.
L 80 I agree with the aim, however, one could suspect that in the aim subsection, the comparison of some varieties / the way of this comparison will be stated as the way to support the first aim thesis. The Sensors reader suspects more numerical than descriptive (L 81-86) phrases.
L 87-89 This sentence should be moved to the beginning of the discussion section.
L 99 Were the heart rate, mucosal and lymph node status, and capillary refill time is taken into account during the initial health assessment of the calves?
What were the criteria for the inclusion of the calf in the group of healthy ones (the threshold/limit for the values of traits assessed in the preliminary examination). Were any calves excluded from the study due to exceeding the limit values? Was the clinical examination repeated before subsequent measurements or during or after the habituation period?
L 142 "spirometry data" or "IOS data"?
L 150 How long did a single measurement last?
It would also be interesting to include RR measurements made by IOS in Figure 3
L 195 How the normal distribution of data was tested?
L 200 and 205 You stated that all tested data series were normally distributed. For such kind of data series, Pearson's correlation coefficient rather than the Spearman’s correlation coefficient is recommended. Hence the question on what basis did you expect AA and why did the potential non-linearity of the correlation influence the choice of the test not recommended for your data series?
L 203 "Since Vt (IOS) is subject to greater fluctuations than the Vmin (IOS)" please justify using numerical data
L 207 How many factors/coefficients have you tested? Can the reader see the results of the tests of better and worse factors/coefficients?
With these changes in mind, I am very much looking forward to reviewing your revised manuscript as it is very valuable for the field cattle practice.
Round 2
Reviewer 1 Report
I wish to thank the Authors for their kind responses to my questions about the initial manuscript.
The new manuscript has been notably improved, and I've gone over the responses with interest, even if the line numbers indicated in the response letter did not match those of the new PDF document provided.
Most of my questions, in my opinion, have been properly addressed by the Authors, but some others seem to have been missed somehow, which I will mention below, using the same numbers for the sake of coherence:
4. In Section 2.1, some explanation about the criteria for selecting the number and age of the animals, the length of the study and other study data is expected. Same can be said for the experiement settings and measurements described in Sections 2.2 and 2.3.
RESPONSE: In Section 2.1 in lines 106 - 107, the length of the study is described (“two consecutive days”). Furthermore, in line 115-117 the age as well as the weight of the calves is described. We have additionally added more details about the selection criteria in lines 110-111:
“Selection criteria for the calf’s inclusion in the testing were clinical health and a maximum weight of 80 kg, since the mask does not fit larger animals.”
In Section 2.2, the length of the measurements is described in lines 158-160:
“The pressure was recorded continuously during inspirations and expirations over a period of 5 minutes per calf and the data were stored on a secure digital memory card.”
In Section 2.3 we described the IOS measurement in more details (lines 173-175):
”Three consecutive measurements were registered per calf per day (each included at least 10 regular breathing cycles free of artifacts) lasting on average 24 s per measurement.”
NOTE: The criteria for selecting the number of the animals, for selecting the length of the study, and justification for the choice of other study data was not provided, nor for the experiment settings and measurements described in Sections 2.2 and 2.3. It is the criteria used what was asked in my 4th review question.
7. Please explain better the meaning of 'decreases hyperbolic with increasing factor' in Line 239. In the same line, elaborate some more the explanation in the description of Figure 9.
RESPONSE: We added the definition of the factor in line 280:
“The characteristic curve shown in Fig. 9 was intended to convert the Vt equivalent values of the RR sensor into true Vt. A power function with hyperbolic course was obtained which means that the Vt equivalent decreases hyperbolic with increasing factor (Vt/Vt equivalent).”
NOTE: In the response provided, and in Line 286 of the revised document, are you referring to 'hyperbolically' when you write 'hyperbolic'?
8. The sentence in Lines 242-243 seriously questions the contributions of this work. I would suggest to address this issue in depth in the Discussion Section.
RESPONSE: Thanks for the advice. The number of animals is often a problem in animal experiments, but the aim of our pilot study was not to create a complete reliable characteristic curve, which would require an extremely large number of animals of different ages, breeds and housing. We mainly wanted to test once how to convert the pressure into a VT equivalent and test it for the first time in calves. In addition, some rework of the RR sensor is needed before a larger animal testing should be performed. I have explained it in more details in the discussion section (lines 349-354):
“When generating the characteristic curve to convert the Vt equivalent into a true Vt, it must be taken into account that the curve is very specific to this age and breed, as it is influenced, among other factors, by the diameter of the nostrils. For an accurate and valid approximation of the calves' Vt across all breeds and ages, it is necessary to expand the characteristic curve based on larger amounts of data with calves differing in age, breed weight and environmental conditions.”
NOTE: All these limitations, in my opinion, indicate that this device is still in a very early research stage. Lessons have been learned by the Authors about how to further improve the system, but it seems to me that it is too early to present these results as meaningful enough to be published in an article.
16. As a consequence of the above considerations, I would think this document needs additional experiments and studies in order to make a relevant contribution to this field of study, and I encourage the Authors to proceed along that way, elaborating and submitting a new version of the manuscript addressing said considerations.
RESPONSE: Thank you for the advice,
The study should primarily show how the pressure can be converted into a Vt equivalent and how this can be converted into a real Vt. It should serve as a basis study for further experiments in this area as a proof-of-concept study and shows the weak points of the RR sensor, which would have to be revised before a further experiment.
NOTE: I appreciate that the Authors are aware of the limited scope of the contributions of this article, and the importance to carry on through this line of work. However, I still think that some more work is to be done to the manuscript to improve its value and relevance. However, I submit my opinions and considerations to the Journal Editor to consider them with a broader view,
Reviewer 2 Report
I appreciate the efforts of the authors to improve the article. Many of my comments have been taken into account correctly, although problems with the statistical analysis of the data still remain.
The statistical analysis section must be rewritten and cannot be accepted in its present form. Therefore, I am asking for a detailed answer to the following question and I am raising still problems in the statistical analysis subsection.
L 236 How was the normal distribution of features tested? Not normal distribution of data can not be used as a reason for calculating the median per calf and per day. Data distribution is the basis for selecting further statistical tests, not recalculating data. Please, rewrite L 235-238.
L 241 You calculate correlation (it is a numerical coefficient) not a relationship.
L 242 Vt equivalent can not behave, as it has no behavior (they are just data)
L 243 On what basis do you expect a nonlinear correlation? The results of the normality test are enough to choose the Pearson correlation coefficient or Spearman correlation coefficient. You don't have to expect anything.
It strongly recommends that authors consult colleagues who are experienced in statistical analysis and revise this subsection. I can offer you the following template. Please correct me if I misunderstand your data set.
You measured the three main parameters (RR, Vt, and Vmin) with two methods (RR sensor and IOS) - yes or no?
You got the data as sequences of numbers as a function of time - yes or no?
If both are yes, you should receive six data series - RR/RR sensor, Vt/RR sensor, Vmin/RR sensor, RR/IOS, Vt/IOS, and Vmin/IOS.
Each data series should be tested independently for data distribution using a normality test (please, provide the name of the used test).
This is your first step. You can show your data as median and quartiles or mean and standard deviation, and it is ok.
In the second step, as it was mentioned more gently in the first round of revision, you should compare the data sets as paired data: 1) RR/RR sensor and RR/IOS, 2) Vt/RR sensor and Vt/IOS, and 3) Vmin/RR sensor and Vmin/IOS (provide the name of the used test - for example, paired t-test when both data series were normally distributed or Wilcoxon matched-pairs signed rank test when one or both data series were not normally distributed as well as the p-value for each comparison).
I know that it requires effort and preparation of new results, but then the presentation of your observations will make sense.
In the third step, you calculated correlation coefficients - The Pearson correlation coefficient when both tested data series were normally distributed or the Spearman correlation coefficient when one or both data series were not normally distributed.
Then clearly specify for which pairs of data you are calculating the correlation coefficient - you can do it for all combinations of data pairs but it makes sense for the following in my opinion: 1) RR/RR sensor and RR/IOS, 2) Vt/RR sensor and Vt/IOS, 3) Vmin/RR sensor and Vmin/IOS, 4) RR/RR sensor and Vt/RR sensor, 5) RR/RR sensor and Vmin /RR sensor, 6) Vt/RR sensor and Vmin /RR sensor, 7) RR/ IOS and Vt/ IOS, 8) RR/ IOS and Vmin / IOS, and 9) Vt/ IOS and Vmin / IOS.
In all those calculations you can use Vt equivalent (RR sensor) values and true Vt values (provide the software for this calculation), thus you should start this section with the introduction of the new variable - true Vt/RR sensor. And repeat all statistical steps also for this variable.
I still argue that the presentation of the results section should be improved to show the results of all the statistical analyzes presented above.
